# The Driving Path of Customer Sustainable Consumption Behaviors in the Context of the Sharing Economy—Based on the Interaction Effect of Customer Signal, Service Provider Signal, and Platform Signal

**Juying Wang \* and Xiaoqing Yu**

Department of Marketing, Managent College, Ocean University of China, 238 Songling Road, Laoshan District, Qingdao 266100, China; 17854257609@163.com
\* Correspondence: wangjuying@ouc.edu.cn

**Abstract:** The sharing economy, based on collaboration, sharing, and innovation, has brought about a disruptive revolution in the transformation of the economy and provided a new operating mechanism for promoting sustainable consumption. Therefore, exploring which signals in the sharing economy can effectively stimulate customer consumption behaviors is of great significance. The research uses the signal-interpretation-response (I-I-R) model to build a research framework for customer sustainable consumption behaviors in the context of the sharing economy. With the help of web crawler technology, we captured customer online review data on Airbnb, the sharing accommodation platform, to study the driving path to interpret how multiple signals from different sources influence sustainable consumption behaviors. Regression research shows that the scores in the customer signal, the sustainable services provided in the service provider signal, the super-host certification in the platform signal, and the interactive effects of the three signals have a significant positive impact on customer sustainable consumption behaviors. Consequently, the increase of customer sustainable consumption behaviors improves sales performance. Furthermore, the fuzzy-set qualitative comparative analysis (fsQCA) found five configurations for customer sustainable consumption behaviors based on different property types. The research results provide a reference for strengthening customer sustainable consumption behaviors and improving the service quality of platforms and service providers.

**Keywords:** sharing economy; customer sustainable consumption behaviors; interaction effect; qualitative comparative analysis

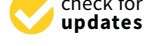

## 1. Introduction

Sustainable consumption behaviors are proposed in the context of sustainable development, the scope of which is broad and considers environmental, economic, and social aspects to reduce waste and energy use and improve well-being in production and consumption processes [1,2]. Sustainable consumption behaviors reflect the significant potential of facilitating the efficient use of underutilized resources and participating in environmental co-governance to form a thrifty, green, and low-carbon lifestyle in all aspects of social life and effectively promote the construction of an ecological civilization [3], both of which coincide with the original intention of the sharing economy [4]. The success of the sharing economy depends on consumers who pay attention to environmental protection interests in pursuing more opportunities for sustainability [5] and evaluate the value of products and services to meet the needs of the world in terms of sustainability [6]. However, the number of these consumers, who tend to prioritize sustainability, is still very small. Meanwhile, There is still an unsustainable phenomenon of wasting, or even destroying, resources in the sharing economy [7]. Therefore, exploring which factors in the sharing economy can effectively stimulate customer sustainable consumption behaviors is not only

a strategy for ensuring sustainability, but also an inevitable requirement for the stable development of sharing economy.

The research on sustainable consumption behaviors in a sharing economy is still in its infancy. Current research only regards sustainability awareness as one of the motivations for sharing accommodations [8–10], and does not explore the promoting mechanism of sustainable consumption behaviors. On the other hand, the network-based and non-contractual sharing economy results in a lack of effective supervision and punishment mechanisms for non-sustainable consumption behaviors [11–13]. Customers can enjoy services without paying additional costs or assets. In order to obtain more benefits, customers who lack environmental motivations will overuse or even destroy shared resources [13], resulting in a waste of resources, which violates the original intention of the sustainable development of the sharing economy. Therefore, this new research content needs to be discussed.

The sharing accommodation has the characteristics of synergy [11,14], and different participants can influence others through the interaction and collaboration of Peer-to-Peer(P2P)networks [14]. Therefore, customer behaviors may be a complex process of mutual influence and interaction between different participants in a variety of consumption scenarios [15,16], including the signals released by platforms, service providers, and early customers to convey their behavior intentions [17]. It is necessary to analyze whether sustainable consumption behaviors are affected by the comprehensive influence of multiple subjects, and whether they vary by product types.

Therefore, this research aims to solve the following questions: In the context of sharing accommodation, which signals affect consumer sustainable consumption behaviors, and do these signals have interactive effects? Secondly, are there different configurations to affect sustainable consumption behaviors based on different property types?

## 2. Literature Review

### 2.1. Sharing Economy and Sustainable Consumption Behavior

Sustainable consumption behavior covers a wide range, including environmental, economic, and social aspects, emphasizing green purchases, resource conservation, waste reduction, and other behaviors to minimize the negative impact of consumption on the environment, or to maintain product quality through voluntary cleaning behaviors [8,18]; however, sustainable consumption behavior requires customers to pay extra care and effort, so it is not easily triggered [19]. The sharing economy, which emphasizes collaborative consumption, can effectively promote the formation and development of customers' sustainable consumption behavior and minimize the use of natural resources and the discharge of waste and pollutants [20]. Because it is different from the traditional economic development model, the sharing economy can more effectively use idle resources instead of wasting too much energy and resources in hotels, for example [21]; at the same time, the sharing and transfer of rights of use can enable buyers to temporarily obtain ownership of a product, and this sense of belonging and autonomy can make customers feel responsible, and thus, minimize the possible adverse effects and consequences of their own behavior, thereby greatly mobilizing customers' enthusiasm and emphasis on participating in sustainable consumption behavior. Customers realize that participating in this consumption model can achieve environmental friendliness, improve resource utilization, and help to achieve sustainable development [22], thereby promoting the formation of sustainable consumption behavior [23].

At present, academic research on sustainable consumption behavior has gradually extended from traditional economic fields (such as green product purchasing, recycling, etc.) to the sharing economy [24–26]. Existing studies have prospectively demonstrated the possibility of achieving sustainable consumption in the sharing economy model [27], such as promoting the development of sustainable consumption in a new economic model and generating new forms of sustainable consumption (such as cleaning, cooking, hygiene, and shopping) [21], or using sustainability as a motivation for customers to participate in the

sharing economy [8]. The antecedent research on sustainable consumption behavior draws on theories of planned behavior in environmental psychology, social psychology, and other disciplines [28], which affect the psychological factors of customers' sustainable consumption behavior. As the entry point for research, the mechanism of action initially formed a research paradigm based on the three levels of individual psychology [29], difference, and social influence [30–32]. For example, research by Sheng and others found that customers' perceptions, care, and attitudes towards the environment significantly affect sustainable consumption willingness and behavior [3,33–36]. However, the sharing economy has synergy characteristics, and the collaboration and interaction between different participants influence each other [14]; on the other hand, sustainability requires common norms and the positive influence of platforms and service providers to increase significantly [37]. Therefore, the customer's sustainable consumption behavior may be affected by the people around them, and the platforms and service providers participating in this behavior are even more critical in the process of the formation of sustainable behavior. It is a kind of customer behavior after a certain degree of knowledge of the economic environment and external influences [38]. Therefore, studying the signals released by platforms, service providers, and customers in the context of the sharing economy is crucial for guiding and supporting customers' sustainable consumption behavior.

### 2.2. Shared Accommodation and Airbnb

As a substitute for traditional accommodation, the emergence of shared accommodation has an important impact on social development. From the perspective of sustainability, shared accommodation enables hosts to transfer the right to use an idle house to obtain economic benefits, and concomitantly reduces the impact on the environment [39]. Therefore, consumers' pursuit of sustainability has given birth to the success of shared accommodation [5]. As the earliest and most extensive short-term rental platform, Airbnb was established in San Francisco, California, USA in August 2008. It is the largest and most popular choice of online travel rental platform.

Airbnb uses excess resources to generate economic benefits to achieve sustainable development [40]. Overall, 72% of guests said that the environmental benefits of Airbnb are one of the reasons why they choose it. Compared with their lives at home, 92% of guests in the United States generate less waste and keep their accommodation clean during their stay. Moreover, 94% of guests recycle items in rented houses. Airbnb guests engage in sustainable development activities every night. Waste is 53% less than that of hotel guests [41]. Chinese homeowners are also playing a role in promoting sustainable tourism. In fact, 95% of homeowners are practicing green hospitality services, such as providing large-capacity toiletries, bus cards, and public transportation reference information [42]. The concept of sustainable consumption has penetrated deeply into the development of Airbnb.

### 2.3. Signal-Interpretation-Response Theory

The signal-interpretation-response (I-I-R) theory evolved on the basis of the signal and signal interpretation research theory of Robertson [43,44]. The signal in this research refers to the information conveying the behavior or opinion of the signal sender [45]. This theory is currently widely used in online shopping situations, because e-commerce platforms are mainly dominated by information technology, which strips out many physical information clues of the product and causes information asymmetry between the sharing parties and the platform [46], as customers usually cannot observe the quality of services or goods before the service or commodity transaction is concluded. At this time, the seller or the platform can transmit product information or behavioral opinions to the customer through observable signals (I) (such as price, introduction, reputation system, etc.), and the customer interprets the received signals (I) to form the preliminary judgment on the platform or service provider, infers the product quality and seller's intention from predictable information [47], and then makes a response (R). Therefore, I-I-R theory can

be used to explain how signals affect customers' perceptions, intentions, and behaviors, and provides a theoretical framework for promoting the formation and development of customers' sustainable consumption behavior.

Previous studies have explored the impact of different signals on consumer behavior from the perspective of signal transmission, such as website quality, product quality and service quality [17], product price, reputation [48], and other quality clues that are related to the wishes and behaviors of customers, but the customer's behavior is ultimately affected by the synergy of multiple signals [49]. In the sharing accommodation platform, due to the different participants, the signals are represented as platform, service provider (that is, the signals conveyed by the homeowner), and customers, as well as their interaction effects. Therefore, based on the theory of I-I-R, this research divides signals into the following three categories: platform signal (super-host certification), service provider signal (provide sustainable services), and customer signal (review scores); the research explores the above three types of signals and their interaction effects on the impact and driving mechanism of customers' sustainable consumption behavior in the context of the sharing economy.

## 3. Research Framework and Hypotheses

According to the theory of social exchange, the customer and the service provider play the roles of both exchanges, and sustainable consumption behavior is considered to be beneficial to the service provider. The sustainable service provided in the service provider signal, the score in the customer signal, and the super-host certification in the platform signal all have an impact on the customer behavior in the sharing accommodation environment and promote an exchange relationship between the two parties [50]. Therefore, favorable signals from different parties can motivate customers to have a beneficiary mentality when they check in and check out, so as to repay sustainable consumption behaviors by keeping the environment clean and reducing waste.

### 3.1. The Impact of Customer Signal on Consumer Sustainable Consumption Behaviors

In the context of online shopping, in order to promote customers' understanding of products, service providers encourage customers to pass product information, related behaviors, and self-perceptions through reputation systems [51], forming user-generated content (UGC) in the form of online reviews and ratings [52], which has an important impact on customer purchasing decisions as an important customer signal [53]. Among them, online scoring is an important measure for judging customer behaviors and opinions [54]. The sharing economy regards keeping products in good condition and keeping a clean environment as sustainable practices [8,18]. Therefore, this study uses the sustainable signals released by customers in ratings as factors that affect customers' sustainable behavior, showing how customers rate clean houses, tidiness, and other sustainable situation ratings (hairy cell and other, 2020). Providers also give customers a rating for the sustainability of the house's cleanliness and tidiness [46]. Customers are more likely to think that high-rated houses are the accumulation of previous customer high ratings, representing a clean and sustainable environment for the listings, and the warm and comfortable feeling of returning home. Therefore, customers are more willing to establish a lasting partnership with high-scoring houses [55] and voluntarily keep the house clean during the check-in process. Accordingly, this study puts forward the following hypothesis:

**Hypothesis 1 (H1).** *The score in the customer signal positively influences the customer sustainable consumption behaviors.*

### 3.2. The Impact of Service Provider Signals on Consumer Sustainable Consumption Behaviors

The influence of society or peers can easily promote customers' sustainable consumption behavior [56], and the influence of service providers, namely homeowners, is particularly important in the practice of sharing accommodation [57]. In order to comply with the strategy of energy saving, emission reduction, and green sustainable development,

the Airbnb community advocates homeowners to provide environmentally friendly and low-carbon products and services to make house more green and sustainable. Therefore, homeowners can provide green and sustainable products and facilities in the introduction of the house or the specific living environment, and create brand-new products and services, so that the concept of sustainable development can not only be integrated into the concept of accommodation, but also penetrate into the behavior of customers. This is because customers may observe and imitate the behavior of the homeowner, and take corresponding energy-saving or cleaning measures to minimize waste and keep the house tidy [58]. In this context, this study defines the homeowner's use of sustainable products or services in the description of housing information or in the physical environment as the homeowner's provision of sustainable services.

Therefore, this research proposes the following hypothesis:

**Hypothesis 2 (H2).** *Service providers (homeowners) providing sustainable services in the service provider signal positively influences customer sustainable consumption behaviors.*

*3.3. The Impact of Platform Signals on Consumer Sustainable Consumption Behaviors*

Most product or service transactions in the sharing economy rely on network platforms and are completed in an anonymous environment. Due to the vagueness of the pre-purchase evaluation of the quality of shared products or services, the network sharing platform participating in the consumption process is at a critical link in the entire chain of behavior occurrence [21]. The "super host" signal released by the sharing platform has become an alternative indicator of product or service quality [54]. A super host is a certification of excellent homeowners. The houses with this title represent a clean and high-quality environment, have continuous consumer orders, and provide professional and friendly services. Therefore, when customers make consumption decisions, they are influenced by awesome homeowners and make sustainable consumption behaviors that are beneficial to the environment and housing. On the basis of this influence, this research proposes the following hypothesis:

**Hypothesis 3 (H3).** *Super-host certification in the platform signal positively influences customer sustainable consumption behaviors.*

*3.4. Interaction Effect*

As mentioned above, customers' sustainable consumption behavior is not driven by one factor, but may be the result of a combination of multiple factors. According to the clue consistency theory, if multiple signals or clues are integrated and consistent, their predictive effects on customer attitudes or behaviors can be added and averaged together [59]. If the early customer scores and platform certification maintain a high degree of consistency, subsequent customers will strengthen the recognition of the sustainable development concept of shared accommodation platforms. This enhanced effect stems from the belief that early customers have the ability to identify sustainable services provided by the homeowner and are likely to carry out the same sustainable consumption behavior [60]. Therefore, the probability of subsequent customers' sustainable consumption behavior is confirmed and strengthened. On this basis, this research proposes the following hypothesis:

**Hypothesis 4 (H4).** *There is an interaction effect between the platform signal, the customer signal, and the homeowners' signal, which comprehensively affect customers' sustainable consumption behavior.*

*3.5. The Result of Consumer Sustainable Consumption Behaviors*

Customers can use the sharing platform to identify the relevant information of house and service providers, use the online reputation system to observe the relevant behaviors of early customers, and thus, make consumption decisions based on this information and improve the intention of reciprocal behavior [61,62]. Therefore, the online review system is an important way to reflect the sustainable consumption behavior of customers, and it is a

positive reference indicator for other customers in evaluating the cleanliness of the house, which can attract a large number of potential customers. Service providers' emphasis on environmental sustainability (that is, the active provision of sustainable services or products) signals that they are environmentalists, and that customers' sustainable consumption behavior (that is, behaviors such as reducing waste or actively cleaning) could be considered as special altruism practices, by making contributions to others and gaining more recognition and respect, which can also enhance the willingness of other customers to pay [63]; this may be particularly true for groups that hold pro-environmental beliefs, and will ultimately be reflected in the growth of sales. Therefore, this research assumes the following:

**Hypothesis 5 (H5).** *Customers' sustainable consumption behavior will have a positive impact on sales performance.*

On the basis of the above assumptions, a conceptual model of the factors influencing customer sustainable consumption behavior based on the I-I-R theory was constructed and is shown in Figure 1.

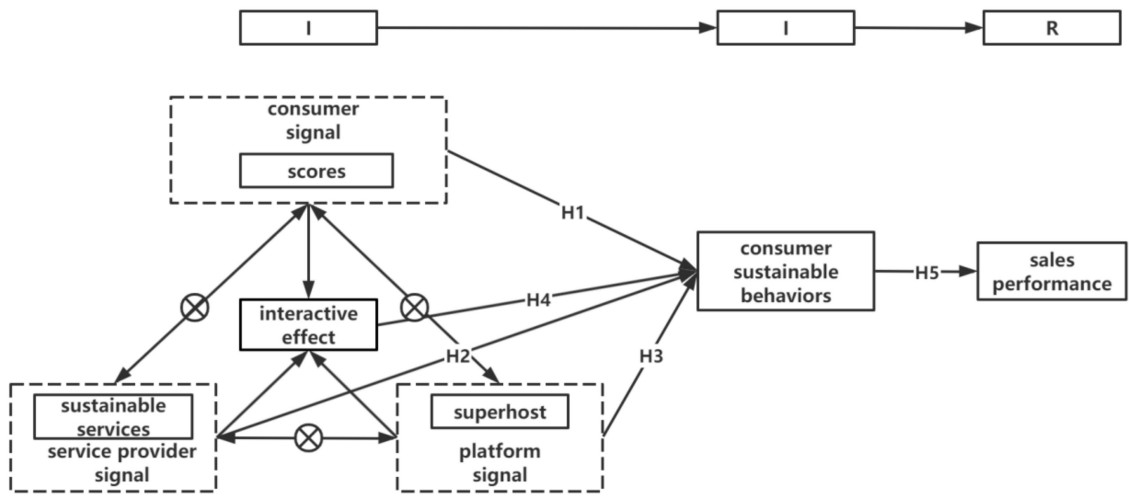

**Figure 1.** Conceptual model of factors affecting sustainable consumption behavior.

## 4. Materials and Methods

### 4.1. Data Preprocessing

We chose Beijing as the sample and retrieved data from Insideairbnb.com for the study. A total of 3279 property data and 53,964 comments were obtained. The number of comments can represent the transaction of listings to a certain extent [61], and the data updated at least once during the year before the download time were shown as active listings [64], so after deleting information that did not meet the requirements and other irrelevant elements, 16,098 comments of 1763 listings remained for the analysis. In the processing of the comment data, we first used Python to delete punctuation marks, numbers, and special characters (#, @, "", / and \, etc.), and used the Jieba tool for word segmentation. As such, the stop words were deleted [6].

Subsequently, it was necessary to filter out those Airbnb users who could be classified as "sustainable consumers" from the reviews. Hurley pointed out that sustainable consumers are those consumers who use words related to sustainable lifestyles [65]. Therefore, words/phrases that specifically mentioned issues related to sustainability, including "green", "responsible for the environment", "environmentally friendly", "organic", and "clean" were selected, which can highlight the behaviors related to sustainable consumption and produce satisfactory results [22,26,40,66–68]. Finally, 942 comments including these

terms were selected. The same content analysis method was adopted when evaluating whether the super host provides a sustainable service.

*4.2. Selection and Measurement of Indicators*

According to research requirements and data processing and index acquisition, the final variables were selected as shown in Table 1.

**Table 1.** Definition and measurement of variables.

| | Variable | Definition | Measurement |
|---|---|---|---|
| Dependent variables | *customer_sustainable_ consumption_behaviors* | Customers support environmental sustainability through voluntary behaviors such as waste reduction and voluntary cleaning [26,69,70] | It was measured by the number words about sustainability mentioned in the customers reviews. [22] |
| | *sales_performance* | The number of orders for the house during the T period [70] | Comment data [61] |
| Independent variables | *sustainable_services* | Whether the homeowner uses green products or services [58] | "1" denotes that the homeowner uses green products or services, "0" denotes otherwise. |
| | *super_host* | Whether the host is certified as a super host. | "1" denotes a super host, "0" denotes otherwise. |
| | *review_score* | A score for "cleanliness and hygiene". | Downloaded from the website |
| Control variables | *distance* | The location of the property [71] | The distance between the house and the city center calculated by latitude and longitude. |
| | *number_of_guests* | Number of customers that the property can hold [22] | The maximum number of customers that the property can hold. |
| | *number_of _rooms* | Number of rooms owned by the property | The maximum number of rooms. |
| | *check_as_ described* | Whether the property is the same as described | The customer's score for the accuracy of the property. |
| | *price* | The price of the property [70] | Price and cleaning fee. |

The descriptive statistics and correlation coefficient analysis of the variables are shown in Table 2. The correlation coefficient of each variable was less than 0.75, indicating that the variables had no serious multicollinearity [72] and that a regression analysis can be performed.

**Table 2.** Descriptive statistics and correlation.

| Varibles | 1 | 2 | 3 | 4 | 5 | 6 | 7 | 8 | 9 | 10 |
|---|---|---|---|---|---|---|---|---|---|---|
| *sales_performance* | 1 | | | | | | | | | |
| *customer_sustainable_ consumption_behaviors* | 0.603 * | 1 | | | | | | | | |
| *sustainable_services* | 0.061 | 0.069 ** | 1 | | | | | | | |
| *super_host* | 0.275 ** | 0.224 * | 0.009 ** | 1 | | | | | | |
| *review_score* | 0.161 | 0.118 ** | −0.023 | 0.217 * | 1 | | | | | |
| *price* | −0.005 * | 0.044 ** | 0.027 | −0.100 | −0.069 | 1 | | | | |
| *distance* | −0.086 ** | −0.041 | −0.077 | −0.029 * | 0.129 | 0.031 * | 1 | | | |
| *number_of_guests* | 0.088 | 0.059 * | 0.089 | 0.073 | 0.274 * | 0.021 | 0.015 * | 1 | | |
| *number_of_rooms* | 0.018 * | −0.014 * | 0.012 ** | 0.002 | 0.544 * | −0.074 * | −0.074 | 0.124 * | 1 | |
| *check_as_ described* | 0.147 | 0.099 * | −0.021 | 0.189 * | −0.041 | 0.675 | 0.687 ** | 0.029 | 0.025 * | 1 |
| mean | 25.847 | 2.968 | 0.264 | 0.219 | 9.637 | 350.36 | 49.944 | 2.499 | 3.226 | 9.667 |
| SD | 32.847 | 5.995 | 0.441 | 0.413 | 0.573 | 198.57 | 17.526 | 1.385 | 2.514 | 0.719 |
| min | 2 | 1 | 0 | 0 | 2 | 47 | 13.003 | 1 | 1 | 2 |
| max | 307 | 134 | 1 | 1 | 10 | 995 | 161.405 | 16 | 16 | 10 |

Note: *: $p < 0.10$; **: $p < 0.05$.

### 4.3. Regression Analysis Results

In order to exclude the influence of extreme values, we censored 99% of each variable to ensure that the dependent variable was normally distributed. Using a multiple regression analysis to test the above hypothesis, the data processing results were obtained and are shown in Table 3.

**Table 3.** Regression results.

| Variables | Customer_Sustainable_Consumption_Behaviors | | | Sales_Performance | |
|---|---|---|---|---|---|
| | **Model1** | **Model2** | **Model3** | **Model4** | **Model5** |
| | Dependent variables | | | | |
| customer_sustainable_consumption_ behaviors | | | | | 0.565 *** |
| | | | | | (0.000) |
| Sustainable_services | | 0.061 *** | 0.035 ** | | 0.014 |
| | | (0.000) | (0.000) | | (0.321) |
| super_host | | 0.216 *** | 0.232 * | | 0.130 *** |
| | | (0.000) | (0.065) | | (0.000) |
| review_score | | 0.101 ** | 0.123 * | | 0.105 *** |
| | | (0.021) | (0.073) | | (0.003) |
| | Control variables | | | | |
| distance | −0.005 *** | −0.002 ** | −0.006 ** | −0.005 *** | −0.004 *** |
| | (0.000) | (0.030) | (0.032) | (0.000) | (0.000) |
| number_of_guests | −0.072 *** | −0.051 ** | −0.061 * | 0.072 *** | −0.050 ** |
| | (0.000) | (0.045) | (0.069) | (0.000) | (0.015) |
| number_of_rooms | −0.053 *** | −0.110 *** | −0.063 *** | −0.053 *** | 0.067 *** |
| | (0.005) | (0.000) | (0.005) | (0.005) | (0.006) |
| checked_as_described | 0.140 *** | 0.028 | 0.156 * | 0.140 *** | 0.053 * |
| | (0.000) | (0.420) | (0.067) | (0.000) | (0.061) |
| price | −0.324 ** | −0.107 *** | −0.268 *** | −0.241 * | −0.044 ** |
| | (0.032) | (0.000) | (0.003) | (0.054) | (0.010) |
| | Interactive effect | | | | |
| Super_host *sustainable services | | | 0.034 ** | | |
| | | | (0.031) | | |
| Super_host *review score | | | −0.296 * | | |
| | | | (0.079) | | |
| sustainable services *review score | | | 0.034 * | | |
| | | | (0.054) | | |
| Super_host *sustainable service *review score | | | 0.345 ** | | |
| | | | (0.043) | | |
| Constant | 0.264 *** | 0.112 * | 0.215 * | 0.263 *** | 0.219 *** |
| | (0.000) | (0.054) | (0.061) | (0.000) | (0.000) |

Note: *: $p < 0.10$; **: $p < 0.05$; ***: $p < 0.01$.

Model 1 in Table 3 introduced all the control variables into the model, while Model 2 added independent variables, such as super-host certification, scoring, and homeowner's sustainability services, to test the influence of tripartite signals on sustainable consumption behavior. It can be seen from Model 2 that the scores in the customer signal ($\beta = 0.101$, $p < 0.05$), the homeowner's sustainability services in the host signal ($\beta = 0.061$, $p < 0.01$), and the super-host certification in the platform signal ($\beta = 0.216$, $p < 0.01$) all had a significant positive impact on customer sustainable consumption behaviors, assuming that H1, H2, and H3 were all supported.

In order to test the interactive effects of the super hosts and homeowners providing sustainable services and scores, the second-order and third-order interaction terms of the above three variables were added to the regression equation. Model 3 showed that the third-order interaction item of *super_host × sustainable_services × review_score* had a significant positive impact on sustainable consumption behavior ($\beta = 0.345$, $p < 0.05$), assuming H4 was supported. When the property was "*super_host × sustainable_services*", high scores

had the strongest positive impact on customer sustainable consumption behaviors; when "non-*super_host* × low *sustainable services*" were considered, the *review score* had a negative impact on sustainable consumption behaviors, and the interactive effects of the three were analyzed, as shown in Figure 2.

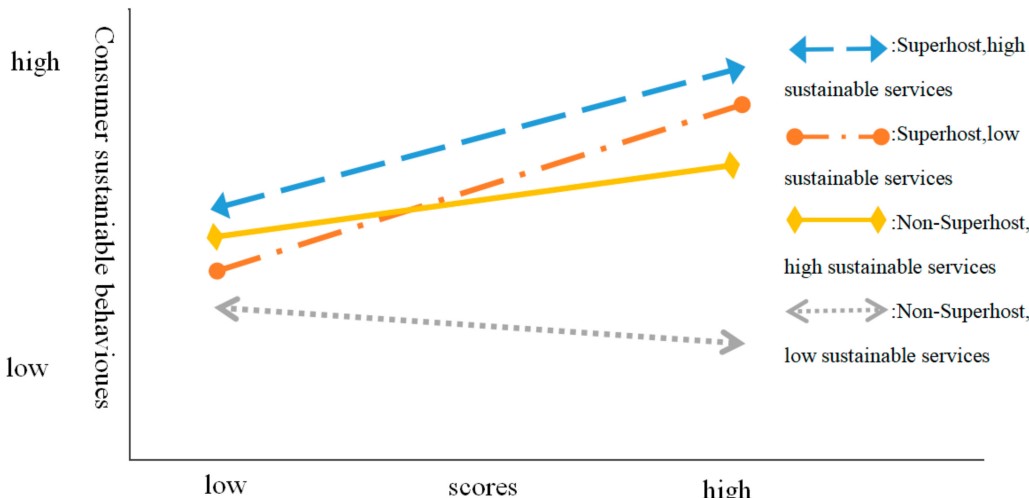

**Figure 2.** The interaction of platform signals, homeowner signals, and customer signals.

Model 4 and Model 5 mainly examined the impact of sustainable consumption behaviors on sales performance. It could be seen from Model 5, shown in Table 3, that sustainable consumption behaviors had a significant positive impact on sales performance ($\beta$ = 0.565, $p < 0.01$); thus, H5 was supported. Emphasis on environmental sustainability establishes a positive image of homeowners, supporting a sustainable consumption environment, and thereby bringing in more passengers and a higher performance.

## 5. FsQCA Analysis of Customer Sustainable Consumption Behaviors

Qualitative comparative analysis is a combination of qualitative and quantitative methods. This method is based on the Boolean algorithm and set idea, which can identify the interactive influence of different antecedent variables and the driving configuration of the result variable. Ragin proposed the fuzzy-set qualitative comparative analysis (fsQCA), which uses a value from 0 to 1 to represent the degree of membership (0 means complete non-membership, 1 means complete membership), avoiding the extreme circumstances that data coding is other than 0 or 1, and maintaining the core principles and operational logic of set theory. In addition, the fsQCA method has low requirements for data. It does not need to set control variables, adjustment variables, and other variables, and does not need to consider the correlation between variables; thus, clear and straightforward analysis results can be obtained. Therefore, we chose fsQCA to study the driving mechanism.

### 5.1. FsQCA Variable Selection and Calibration

The main variables and the intermediate variables involved in the study that significantly affected the outcome variables were selected as the antecedent variables, that is, homeowners providing sustainable services, super-host certification, price, scores, distance, number of guests, and number of rooms. Considering that different room types may have different impacts on customer sustainable consumption behaviors, the data were divided into whole houses, independent rooms, and shared rooms. The coordinated linkage of dependent variables is the driving path for customer sustainable consumption behaviors in different product consumption situations.

The premise of fsQCA analysis is to calibrate the antecedent variables in the study to improve the interpretability of the results. With reference to previous research, this paper set the three anchor points of all the variables according to the upper quartile, median, and

lower quartile of the sample data [73]. After the data were calibrated, it was necessary to analyze the necessity and sufficiency of each variable, and it was found that the antecedent conditions of each variable did not meet the necessary condition standard; that is, the coverage value was less than 1 and the antecedent conditions were not sufficient; therefore, it was necessary for this study to analyze the combined configuration of each antecedent variable. The specific results are shown in Table 4.

**Table 4.** Variable Necessity and Adequacy Test.

| Variables | Consistency | Coverage |
|---|---|---|
| *sustainable_services* | 0.289 | 0.504 |
| *~sustainable_services* | 0.711 | 0.458 |
| *super_host* | 0.320 | 0.679 |
| *~super_host* | 0.680 | 0.411 |
| *review_score* | 0.591 | 0.584 |
| *~review_score* | 0.545 | 0.489 |
| *distance* | 0.518 | 0.509 |
| *~distance* | 0.598 | 0.539 |
| *number_of_guests* | 0.739 | 0.602 |
| *~number_of_guests* | 0.548 | 0.610 |
| *number_of_rooms* | 0.785 | 0.536 |
| *~number_of_rooms* | 0.429 | 0.647 |
| *price* | 0.607 | 0.509 |
| *~price* | 0.510 | 0.545 |

*5.2. Empirical Results of FsQCA*

Using the fsQCA analysis software developed by Ragin, the consistency threshold was set to 0.8, and the case threshold was set to 1. The final configuration of customer sustainable consumption behaviors based on the different house types is shown in Table 5.

**Table 5.** The fsQCA analysis results of customers' sustainable consumption behavior.

| Configurations | W (Whole House) | | P (Private House) | | S (Shared House) |
|---|---|---|---|---|---|
| | W1:Economy-Driven | W2:Full-Process-Driven | P1:Reputation-Driven | P2:Quality-Driven | S1:Low-Price-Driven |
| *sustainable_services* | | ● | ● | • | • | • |
| *super_host* | | ● | ● | ● | ● | ● | • |
| *price* | ⊕ | ⊕ | | ○ | | ● | ⊕ |
| *review_score* | • | | ● | ● | ● | | • |
| *distance* | | ○ | ● | | • | • | ⊕ |
| *number_of_guests* | • | | | • | | | |
| *number_of_rooms* | | • | • | • | • | | ○ |
| CS | 0.907 | 0.854 | 0.988 | 0.841 | 0.853 | 0.809 | 0.814 |
| CV | 0.410 | 0.251 | 0.058 | 0.190 | 0.361 | 0.308 | 0.342 |
| NCV | 0.041 | 0.0218 | 0.026 | 0.141 | 0.127 | 0.242 | 0.245 |
| OCS | | 0.889 | | | 0.832 | | 0.654 |
| OCV | | 0.835 | | | 0.518 | | 0.786 |

Note: (1) ●and •indicate that the condition exists; ⊕ and ○ indicate that the condition does not exist; "blank" indicates that the condition may or may not exist in the configuration; (2) ●or ⊕ indicates the core condition, •or ○ indicates the auxiliary condition; (3) CS stands for consistency, CV stands for coverage, NCV stands for net coverage, OCS stands for overall consistency, and OCV stands for overall coverage.

According to the QCA analysis results, the following configurations were obtained:

- The whole house
  1. Economy-driven type: The core condition of the W1 configuration included *~price*, and *super_host* also appeared as a core condition in this configuration. It showed that low prices can effectively promote customer sustainable con-

sumption behaviors. This is because low prices can reduce the risk of customers' economic losses, thereby prompting customers to reward economic transfer through sustainable behaviors. Therefore, for homeowners who are committed to attracting customers at low prices, they need to strengthen the professionalism of services and maintain the continuity of orders. At the same time, in terms of sustainability, they should respond to the sustainability policies proposed by the platform or the government, provide sustainable solutions in line with the characteristics of the house, and take the initiative to convey green and low-carbon views and behaviors to customers through communication with customers, reflecting the sustainability of their products.

2. Full-process-driven type: The core conditions of the W2 configuration included *sustainable_services, super_host, review_score,* and *distance,* indicating that many factors affect the sustainable consumption behavior of customers, including sustainable services, professionalism, higher scores, and the location away from the city center. Among them, the geographical location can be interpreted as the property is located on the outskirts of the center, with an incompletely urbanized outdoor environment. The original ecological environment encourages customers to maintain the unity and coordination of indoor hygiene and the outdoor environment. For such customers, the homeowner should attack in an all-around way, letting the concept of sustainable consumption be reflected in all aspects of the consumption process, which will have a subtle impact on customers. They should also place indoor facilities and products according to the environment of the house to fully reflect the low-carbon, green, and sustainable atmosphere.

- Private house

1. Reputation-driven type: The core conditions of the P1 configuration included *super_host* and *review_score,* which indicates that the high reputation of super-hosts and scores can bring a high probability of sustainable consumption behavior, which fully reflects the power of reputation and word of mouth. This type of homeowner can consider using the reputation system launched by the platform and work hard on customer feedback after purchase. The homeowner can attract customers to make clear and positive comments on the sustainability of the house and the homeowner through rewards or gifts. The publicity of word-of-mouth reviews influences subsequent customers to make more sustainable consumption behaviors that also clean up and reduce waste; on the other hand, the platform can design a corresponding display or reward system based on the sustainable behavior of homeowners and customers to encourage the proactiveness of buyers and sellers.

2. Quality-driven type: The core conditions of the P2 configuration included *super_host* and *price,* which indicates that a high-quality, high-priced property owned by a super host will bring sustainable behavior. Comparing P2 with W1, it can be seen that *distance* appeared in P2 as an auxiliary condition. This may be explained by the fact that the houses in P2 were independent houses with higher overall quality and were far from the city center, meeting the needs of customers for quiet, clean, and small houses. For such customers, exquisiteness, independence, and cleanliness can meet their needs. Moreover, such customers may have relatively high levels of education and lifestyle. Therefore, they have a more independent and comprehensive understanding of environmental protection and waste reduction. Therefore, on this basis, homeowners can maintain a clean living environment and use eye-catching green products to stimulate customer sustainable consumption behaviors.

- Shared house

Low-price-driven type: The core conditions of the S1 configuration were *~price* and *~distance*, indicating that low-cost shared houses close to the city center can promote sustainable consumption behaviors. Combining the characteristics of shared houses, this type generally refers to economical and practical youth hostels or dormitory houses, with simple product facilities and fewer public resources, so the waste of resources and environmental damage is reduced accordingly. Customers may be pressured by people around them to minimize the occurrence of unsustainable behaviors for the purpose of maintaining a clean and tidy public living environment and maintaining a good image of themselves. Homeowners can set up eye-catching slogans to protect the environment and reduce waste in public areas, and conduct regular inspections and reminders to urge customers to respond actively and autonomously.

## 6. Conclusions and Suggestions

This article is mainly based on the signal-interpretation-response theory, with the shared accommodation platform Airbnb as the research object, and crawler data as the support, using multiple regression methods to explore the effect of platform signals, customer signals, service provider signals, and the interaction of the three in the context of the sharing economy, on the customer sustainable consumption behaviors. The results show that the super-host certification in the platform signal, the homeowners providing sustainable services in the service provider signal, and the scores in the customer signal have a positive effect on customer sustainable consumption behaviors. In addition, using fsQCA to further derive five antecedent configurations based on different product types, namely, the economy-driven type, full-process-driven type, reputation-driven type, quality-driven type, and low-price-driven type. These configurations provide an important reference for the sharing economy platform and service providers to improve their services and implement the sustainable development goals.

### 6.1. Conclusions

First of all, this research can provide a certain supplement to customer sustainable consumption behaviors based on the sharing economy background. A series of previous research experiences emphasized customer sustainable consumption behaviors in the traditional economy. However, there is less research on the factors affecting customer sustainable consumption behaviors in the context of the sharing economy, and there is a lack of research from the perspective of the synergistic effect of multiple participants to study the comprehensive driving path of multi-signal effects on customer sustainable consumption behaviors.

Secondly, we expanded the use of research perspectives and methods. Previous research focused on taking the subjective perception and behavioral willingness of customers as dependent variables, and obtained data in the form of questionnaires, lacking a certain degree of objectivity, and rarely combining housing data with review data to objectively explore the customer consumption behaviors. We used crawler technology to obtain objective data to explore sustainable consumption behaviors in the sharing economy and determine indicators such as platform signals, customer signals, and service provider signals. We used a combination of qualitative and quantitative methods to explore customer sustainable consumption behaviors, which makes up for previous research with data subjectivity and lacking a single source.

Finally, by exploring customer sustainable consumption behavior as a reciprocal behavior in the sharing economy, the literature on social exchange theory was enriched. This study found that customers reflect the concept of reciprocity in social exchange theory through sustainable consumption behaviors in the exchange process. Service providers provide high-quality services, high-quality facilities, and a clean environment, while customers reduce waste and keep the environment clean during the staying process.

*6.2. Suggestion*

The research provides some valuable insights for sharing platforms and service providers to further guide and improve customer sustainable consumption behaviors.

- Platform

  1. Platform rules related to sustainable consumption behaviors should be set up, such as setting up push messages to remind customers to keep the environment clean and tidy during their stay and to generally reduce waste.

  2. The platform should improve customer evaluation and feedback mechanisms by providing certification labels such as "green" and "sustainability" in the certification or review mechanism for homeowners, and by encouraging homeowners and consumers to actively demonstrate the relevant information about sustainable consumption behaviors, awakening customers' awareness of sustainable consumption through visual text.

  3. The platform should set up incentive policies to encourage customer sustainable behaviors and homeowner sustainable behaviors. Platform administrators can give economic or point rewards to homeowners who provide environmentally sustainable products and customers who highlight sustainable consumption trends in their reviews. They could also provide price concessions for platform products or related services.

  4. The platform should reasonably use online marketing channels to strengthen the advocacy and publicity of sustainable consumption behaviors, establish shared interest communities, actively encourage customers to participate in online and offline activities, and provide a good atmosphere for customers to participate in sustainable consumption behaviors.

- Service providers

  1. They should comprehensively understand and recognize customer needs and characteristics based on different product types, so as to "prescribe the right medicine" for their characteristics, cut into services from different aspects, and have a targeted and focused product, environmental, and professional service.

  2. They should increase investment in sustainable products and services, use sustainable products as much as possible, and reflect on the use of such products in personal introductions, so that customers can have a real experience and feelings in the consumption process, thereby driving sustainable consumption behavior.

  3. Effective use of the reputation system is important. Service providers can encourage customers to evaluate and rate the sustainability of products after consumption, thereby enhancing the influence of user-generated signals on sustainable consumption behavior.

  4. They should actively strive to obtain relevant certification of the platform for sustainable products, and realize the importance of the influence of platform endorsement on customer behavior.

*6.3. Limitations*

This study also has the following limitations:

1. The research object selected in this article is the specific shared accommodation platform Airbnb, but given that the sharing economy has multiple types of operations and models, future research can be extended to other shared economy fields or accommodation platforms. Future research should carry out horizontal and vertical comparisons to test the influence paths of signals based on different platform backgrounds on customer sustainable consumption behaviors.

2. The research data were limited to Beijing, China, mainly because the InsideAirbnb.Com website only provides information in mainland China. Beijing's data has limitations on the universality of research results. In the future, we can study whether different countries and cultural backgrounds will affect customer sustainable consumption behaviors.

3. Future research can also expand data sources and use interview or questionnaire data to further verify the research conclusions.

## 7. Patents

There are no patents resulting from this study.

**Author Contributions:** The contribution of all authors was balanced in all phases of the development of this study, both in the empirical part (creation and validation of the instrument, data collection, and analysis) and in the writing part of this manuscript and its various parts. Even the writing of the discussion and the conclusions was produced from a debate among the contributors of the work, which allowed for enriching the arguments based on the different opinions presented. All authors have read and agreed to the published version of the manuscript.

**Funding:** This research was funded by Humanities and Social Science Youth Program of the Ministry of Education of China, grant number 16YJC790099, and Philosophy and Social Science Planning Project of Shandong Province of China, grant number 15CGLJ34.

**Institutional Review Board Statement:** Ethical review and approval were waived for this study, as it did not involve personally identifiable or sensitive data.

**Informed Consent Statement:** The study do not involve human nor sensitive data.

**Data Availability Statement:** The data presented in this study are available on request from the corresponding author or directly download from www.insideairbnb.com (accessed on 20 April 2020).

**Conflicts of Interest:** The authors declare no conflict of interest.

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
