# Peer review of "The Driving Path of Customer Sustainable Consumption Behaviors in the Context of the Sharing Economy—Based on the Interaction Effect of Customer Signal, Service Provider Signal, and Platform Signal"

_sustainability, doi:10.3390/su13073826_

Round 1

Reviewer 1 Report

The paper is interesting;

however it is necessary to improve the method section...it is not clear.

also the tables with variables name are hard to understand.

In addition, the English is so poor that reading this paper will be like swimming in porridge.

Author Response

Firstly,I have supplemented the QCA research method, which is a combination of qualitative and quantitative methods.Secondly,I have explained the meaning of variables in detail and modified in the table.Meanwhile,the English expression has been improved.Please refer to the paper for details.
Thanks  for your valuable comments.

Reviewer 2 Report

The article presents a framework for customer sustainable consumption behaviors in the context of Airbnb.  Review comments available on InsideAirbnb were extracted and analyzed. Overall, the methodology, based on the theory of social exchange and the web scraping method with phyton, is technically sound. I applaud the authors for the sound methodology used in this study. This topic has some pragmatic value and it is worth investigating. However, there are several major issues in this work that need to be further improved, 

1. Literature Review: Please also review past studies on green consumerism in the lodging sector and relates it to Airbnb. Current references to research works that are along the line research of this article are missing, in which the behaviour of sustainable consumption, the web scraping technique and the collaborative economy are related, such as that of Serrano, Ariza-Montes, Nader, Sianes and Law (2020), Cheng and Jin (2019), or Luo and Tang (2019).

2. My main concerns are with the data collection criteria and their selection. One of the disconnected parts in this work is that, the analysis was based on several online reviews related to some selection variables. How can we ensure that those comments are from the targeted group of consumer with a sustainable consumption behaviour?, It is not to reduce too much the concept of sustainable consumption behaviour (variable) to the definition offered by the authors and which is even more reductionist when they select the comments related to this variable according to two words selected and related to cleaning?, If only a small part of comments of the potential target is analyzed, the conclusions might not be generalized well.

3. In addition, some explanation on how a data preprocessing step to clean the text details seem missing from the paper. Did the author apply any text processing operation (e.g. word normalization to lower cases, symbol removal to keep only characters, stop word removal, long/short word removal, part of speech tagging)?

4. The information in the table 2, table 3,table 4 and table 5, in the column of the “variable”, It is not entirely clear, perhaps it is better to put the name of the integer variable for more information for the reader.  

Author Response

Your valuable comments have provided a great help in improving my article, so I amended it.

First, I added research on sustainable consumption behaviors of Airbnb in  Literature Review.

Second, about data collection criteria and their selection. I referred to Serrano, Ariza-Montes, Nader, Sianes and Law (2020), Cheng and Jin (2019), Wang (2019, and Luo and Tang (2019)'s article,and filtered sustainable consumers based on words such as "sustainability", "organic", and "green" in numerous review data, and studied the influence of signals on them

About data preprocessing step to clean the text details, I also refer to the processing methods of the above scholars to clean the comment data, including removing stop words and punctuation, using the jieba system for word segmentation, etc.

Finally, I have put the name of the integer variable in the table 2, table 3,table 4 and table 5 for more information.

Please refer to the paper for details. Thanks again for your comments.

Reviewer 3 Report

The article deals with current topic of customer sustainable consumption behaviors in the context of the sharing economy. The article is built on a robust theoretical background. The authors used adequate mathematical and statistical methods that positively contribute to the originality of the articles. The results of the work contain high added value and are relevant for further research in the field of customer sustainable consumption behaviour. The argumentation and findings are coherent. The entire sturcture of the research paper is well-balanced. The authors clearly defined the limits of the research. The final section succinctly summarises the results of the research. I consider the article to be high quality and recommend it for publication.

Author Response

Thanks  for your valuable comments and recognition of this article, I will continue to improve my paper until it is published.Have a good day!

Round 2

Reviewer 1 Report

The actual paper version is wel done. The authors have followed my suggestions.